# An instant messaging mobile phone application for promoting HIV pre-exposure prophylaxis uptake among Chinese gay, bisexual and other men who have sex with men: A mixed methods feasibility and piloting randomized controlled trial study

**Chunyan Li**[1,2,3], **Yuan Xiong**[3], **Suzanne Maman**[1], **Derrick D. Matthews**[1], **Edwin B. Fisher**[1], **Weiming Tang**[3,4], **Haojie Huang**[5], **Tong Mu**[6], **Xiaokai Tong**[7], **Jianxiong Yu**[8], **Zeyu Yang**[3], **Renslow Sherer**[9], **Aniruddha Hazra**[9], **Jonathan Lio**[9], **Linghua Li**[10], **Joseph D. Tucker**[3,11,12] *, **Kathryn E. Muessig**[1]

**1** Department of Health Behavior, Gillings School of Global Public Health, University of North Carolina at Chapel Hill, Chapel Hill, North Carolina, United States of America, **2** Tokyo College, The University of Tokyo, Tokyo, Japan, **3** University of North Carolina at Chapel Hill, Project-China, Guangzhou, Guangdong, China, **4** Dermatology Hospital of Southern Medical University, Guangzhou, Guangdong, China, **5** Wuhan Tongxing LGBTQ Center, Wuhan, Hubei, China, **6** Qingdao Eighth People's Hospital, Qingdao, Shandong, China, **7** Xi'an Polytechnic University, Xi'an, Shannxi, China, **8** Beijing Naomi Media Company, Beijing, China, **9** Department of Medicine, University of Chicago, Chicago, Illinois, United States of America, **10** Department of Infectious Diseases, Guangzhou Number Eight People's Hospital, Guangzhou, China, **11** Institute of Global Health and Infectious Diseases, University of North Carolina at Chapel Hill, Chapel Hill, North Carolina, United States of America, **12** London School of Hygiene and Tropical Medicine, London, United Kingdom

* jdtucker@med.unc.edu

## Abstract

### Background

Mobile health (mHealth) is a promising intervention mode for HIV prevention, but little is known about its feasibility and effects in promoting pre-exposure prophylaxis (PrEP) uptake among Chinese gay, bisexual and other men who have sex with men (GBMSM).

### Methods

We evaluated an instant messaging application using a WeChat-based mini-app to promote PrEP uptake among GBMSM via a mixed-methods design that includes a 12-week, two-arm randomized controlled pilot trial and in-depth progress interviews in Guangzhou, China. Primary outcomes include the number of PrEP initiations, individual-level psychosocial variables related to PrEP initiation, and usability of the PrEP mini-app.

### Results

Between November 2020 and April 2021, 70 GBMSM were successfully enrolled and randomized into two arms at 2:1 ratio (46 to the intervention arm, 24 to the control arm). By the

**Data Availability Statement:** We cannot share the study data to researchers outside of the research team without participants' written permission, as it contains sensitive patient information (e.g., HIV status, medical prescription, etc). This was explained explicitly in the participants informed consent form. Any request for the data should be addressed to the corresponding author Dr. Joseph D. Tucker or by email to jdtucker@med.unc.edu; or the UNC Institutional Review Board at +001-919-966-3113 or by email to IRB_subjects@unc.edu; or contact the Guangzhou Eighth People's Hospital Institutional Review Board at +86-020-83838688.

**Funding:** This study was supported by the National Institute of Allergies and Infectious Diseases (Grant# R01-AI114310-S1, JT), National Institute of Mental Health (Grant # R34119963, JT & WT), National Natural Science Foundation of China (Grant#81903371, WT), the Graduate School of University of North Carolina at Chapel Hill (CL), and Tokyo College at the University of Tokyo (CL). The funders had no role in study design, data collection and analysis, decision to publish, or preparation of the manuscript.

**Competing interests:** The authors have declared that no competing interests exist.

end of 12-week follow-up, 22 (31.4%) participants completed the initial consultation and lab tests for PrEP, and 13 (18.6%) filled their initial PrEP prescription. We observed modest but non-significant improvements in participants' intention to use PrEP, actual PrEP initiation, PrEP-related self-efficacy, stigma, and attitudes over 12 weeks when comparing the mini-app and the control arms. Qualitative interviews revealed the key barriers to PrEP uptake include anticipated stigma and discrimination in clinical settings, burden of PrEP care, and limited operating hours of the PrEP clinic. In-person clinic navigation support was highly valued.

## Conclusions

This pilot trial of a mobile phone-based PrEP mini-app demonstrated feasibility and identified limitations in facilitating PrEP uptake among Chinese GBMSM. Future improvements may include diversifying the content presentation in engaging media formats, adding user engagement features, and providing off-line in-clinic navigation support during initial PrEP visit. More efforts are needed to understand optimal strategies to identify and implement alternative PrEP provision models especially in highly stigmatized settings with diverse needs.

## Trial registration

**Trial registration**: The study was prospectively registered on clinicaltrials.gov (NCT04426656) on 11 June, 2020.

## Introduction

Oral antiretroviral medicine-based pre-exposure prophylaxis (PrEP) for HIV prevention has been widely acknowledged as a safe and effective tool for HIV prevention among gay, bisexual and other men who have sex with men (GBMSM) [1]. The China National Medical Products Administration (formerly the State Food and Drug Administration) approved the combination of emtricitabine (FTC, 200 mg) and tenofovir disoproxil fumarate (TDF, 300 mg) for PrEP in August 2020 [2] and further approved China-produced generic TDF-FTC that significantly reduced the financial cost from 1980 Chinese Yuan/Month (~305 USD/month) to as low as 350 Chinese Yuan/month (~52 USD/month) in 2021. However, PrEP is still not readily accessible to the majority due to the country-wide limited capacity of PrEP provision [3], as formal PrEP provision is mostly centralized in infectious diseases hospitals where HIV-related clinics are often located in non-obvious areas.

While previous studies have indicated PrEP is highly acceptable among Chinese GBMSM [4], fewer than 6,000 people in China were currently using PrEP in April 2021 [5]. In 2020, about 30% of the new HIV infections in China are transmitted via male-to-male sex activities, and in some major cities GBMSM account for more than 75% of all new HIV infections [6]. The large and diverse population base in China who would benefit from PrEP poses significant challenges for timely PrEP scale-up across the country. The ubiquitous use of mobile phones for health information seeking and communication among Chinese populations has made mobile phones an ideal platform for HIV/sexual health-related interventions, including PrEP promotion. Previous studies in global settings have revealed success of mobile health (mHealth) in facilitating PrEP scale-up by removing the barrier of physical distance to PrEP

clinics through video consultation and mailing services for lab tests and medications, increasing care adherence through real-time electronic reminders, enhancing patient education through rich media content [7–11]. In Chinese populations, mHealth-enabled intervention studies have also proven successful in increasing HIV self-testing, reducing risk behaviors, and improving mental health status among people living with HIV [12–14]. However, to date, there is little evidence of theory-informed mHealth behavioral intervention studies for facilitating PrEP uptake among GBMSM in China.

Informed by intensive formative research on mHealth-based intervention studies for HIV prevention among Chinese GBMSM [15,16] and the products from a Gay-friendly Doctor Finder Hackathon [17], we developed a WeChat-based PrEP education mini-app tailored for Chinese GBMSM in the Guangzhou area at substantial risk of HIV infection [18]. Health "hackathons" are an effective and convenient crowdsourcing approach to mobilize GBMSM communities in generating mHealth solutions that meet their own health needs and priorities [19]. In this paper, we reported the feasibility and preliminary efficacy results of the PrEP mini-app from a pilot two-arm randomized controlled trial (RCT) with a small sample of Chinese GBMSM living in the Guangzhou area. Using both qualitative and quantitative approaches, we assessed how individuals' behavioral determinants of PrEP initiation (e.g., PrEP use self-efficacy, stigma, and the stages of behavioral change towards PrEP initiation) and PrEP initiation might change with exposure to the mini-app. The findings from the study are reported in this paper to help fill the gap in our knowledge of developing culturally appropriate and GBMSM-friendly interventions. This information can be used to prepare GBMSM for PrEP uptake in China and guide further refinements to this intervention tool to better meet GBMSM's needs for broader implementation.

## Methods

### Study overview

The PrEP mini-app pilot study was a two-arm RCT with 70 self-reported HIV-negative GBMSM who lived in the Guangzhou area, China during the study period (November 2020-April 2021). The study lasted 12 weeks, where the first eight weeks were the active intervention period and the last 4 weeks were post-intervention observation. Fig 1 depicts the CONSORT flow diagram for this study. The trial protocol that reports details of the study design, intervention development, participant recruitment, inclusion criteria, screening, and randomization process can be found in the supplementary information [18]. In brief, participants were recruited through online social media advertisements and referrals by local LGBTQ community-based organizations in Guangzhou. Potential participants were invited to complete a virtual enrollment visit via a WeChat video/audio call for eligibility screening and informed consent process. Eligible participants were immediately enrolled after screening and consent, instructed to complete a self-administered Web-based baseline survey, and randomly assigned to the mini-app group (i.e. the intervention app plus standard care) or the control group (i.e. standard care only) at a 2:1 ratio using a permuted block randomization. The 2:1 allocation was used to ensure the intervention arm had enough samples to capture the range of users' reactions to the mini-app and its content. Randomization sequence was created using Stata 15.0 (StataCorp LLC. College Station, TX) with block size of six. Both research staff and participants were not blinded to their assigned study arm condition due to the nature of the intervention.

**Intervention: The PrEP Mini-app.** The primary participant-facing component of the intervention was the PrEP education WeChat-based mini-app (Fig 2). Available both in Android and iOS mobile devices, WeChat is one of the most common and important virtual

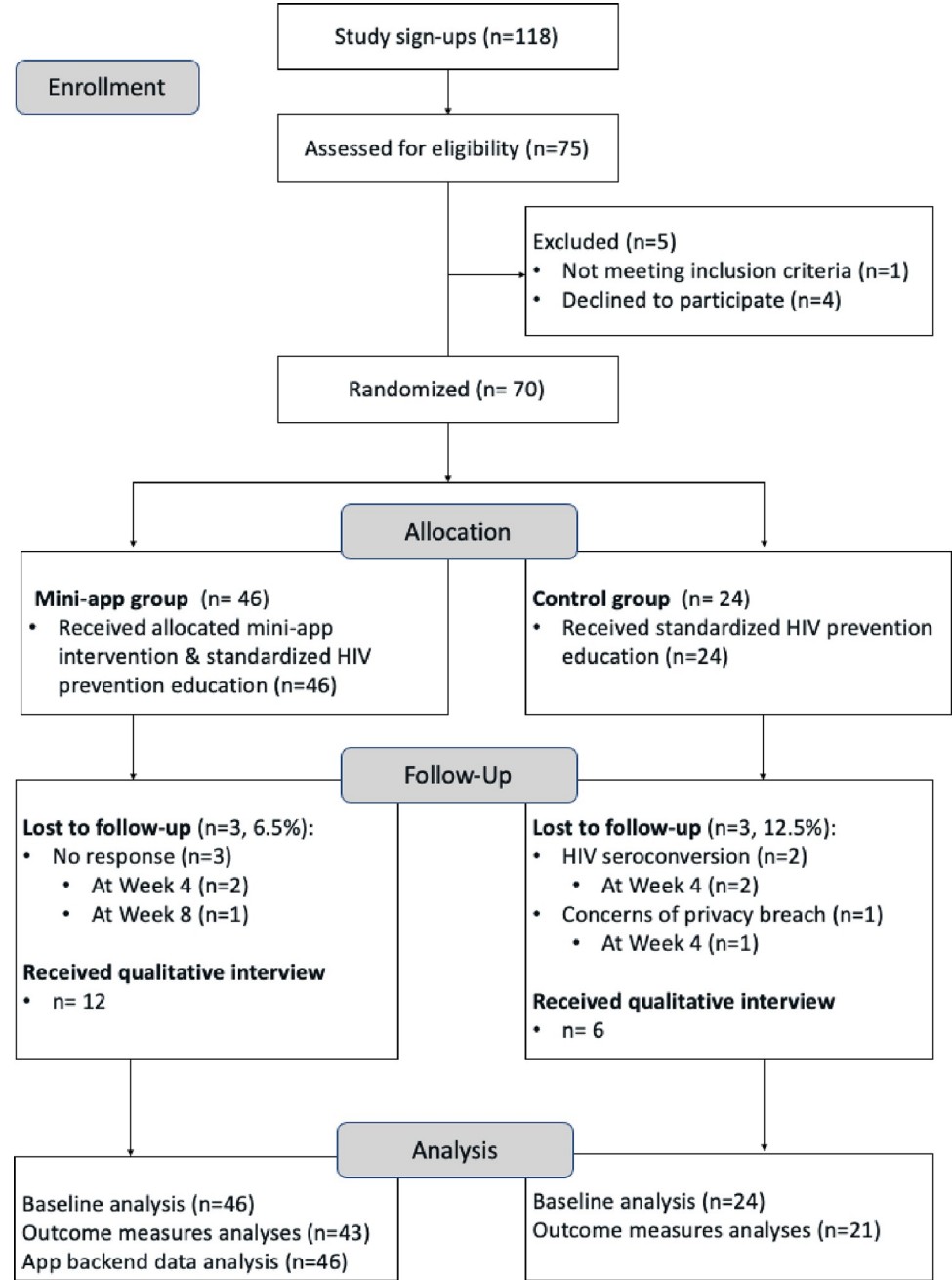

**Fig 1. CONSORT flow diagram of participants in the two-arm PrEP mini-app RCT.**

social media and multipurpose applications in China. WeChat allows third-parties to develop their own programs providing features to users within the WeChat app, i.e. "a sub-app within an app", or the mini-app. WeChat mini-apps are known for their light-weight in terms of phone storage, instant loading, potential to reach millions of active WeChat users, and not requiring additional downloads or sign-ups [20].

The initial prototype of the PrEP education mini-app was developed through a Gay-Friendly Doctor Finder Hackathon [17], and was refined through two phases before testing in this RCT. The first phase was a co-development process guided by the Information-

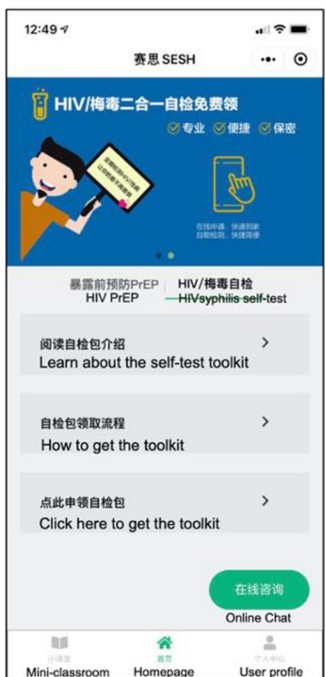
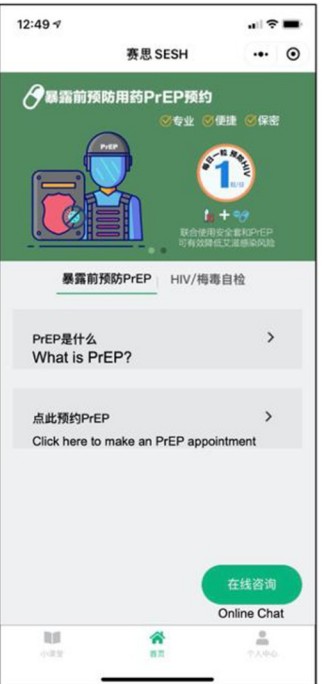
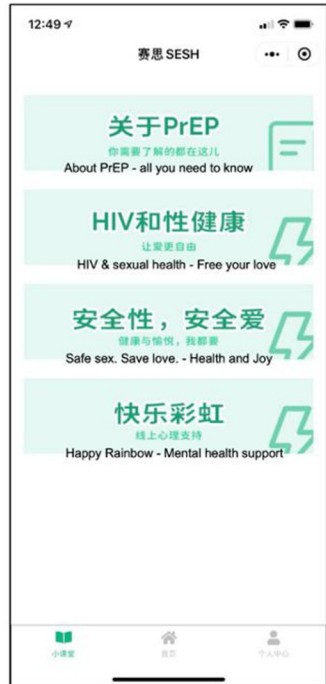
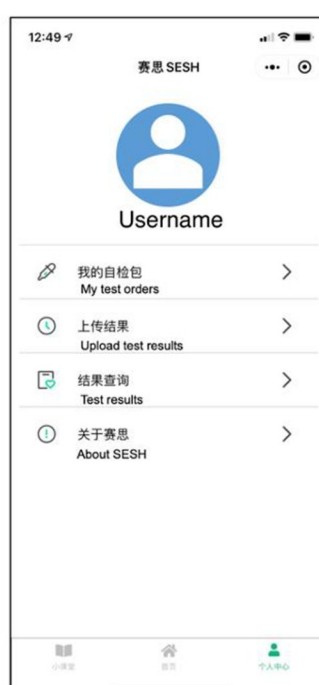

**Fig 2. Screenshots of the PrEP mini-app.** The original mini-app interface is in simplified Chinese only. English translation is provided here for international readers' interpretation. Reprinted from Li et all [18] under a CC BY-NC 4.0 license, with permission from BMJ Open, original copyright 2021.

Motivation-and-Behavioral Skills Model (IMB) where the research team worked with community partners, HIV physicians, and sex education experts to develop the intervention contents and user-interface design. Following this, we recruited 31 GBMSM to test use the mini-app for 5 to 10 minutes and give verbal feedback during virtual one-to-one in-depth interviews. We then further refined the mini-app based on their qualitative feedback. The finalized PrEP mini-app has four key features corresponding to theoretical constructs of the IMB model (details presented in S1 Table), including: (1) Mini-classroom: 59 articles in Chinese on four themes: PrEP, HIV and sexual health, safe sex and mental health; (2) Online chat: asynchronous private chat box monitored and responded by CL; (3) HIV/syphilis self-test kit ordering system that provided free self-test kits via mail delivery, and (4) User profile center to allow participants to keep track of their HIV/syphilis testing behaviors.

The PrEP mini-app was password-protected to restrict access solely to participants in the intervention arm. The mini-app was able to collect the aggregate numbers of daily visits, but was not able to track individual user information or activity. Use of the mini-app was at participants' discretion or preference. However, the study team sent weekly reminders via WeChat text messages that encouraged participants to use the mini-app. After Week 8, participants in the intervention arm no longer received reminder messages but could continue using the mini-app throughout the remainder of the study period.

**Control condition: Standardized HIV prevention education.** Participants in both groups were offered standardized HIV prevention and testing education during their initial and final study visits, including printed or electronic HIV prevention materials about PrEP and HIV/STI testing, referrals to local prevention services, and a description of the standard procedures to access PrEP through the study-affiliated hospital.

**PrEP initiation and follow-up.** As described in the trial protocol [18], participants in both groups could choose to initiate PrEP through the study at any time point from enrollment

through the end of the eighth-week post-enrollment. As the study enrollment visit was completely remote, an additional in-person clinic visit was required for participants who planned to initiate PrEP after study enrollment. During the initial clinical visit, a local research assistant would walk the participant through the registration and laboratory test process at the study hospital.

The expense of physical examinations (including required lab tests), PrEP prescriptions and PrEP medications was partially covered by the study. Participants paid for their PrEP medications 100% out-of-pocket up-front, and then submitted a copy of their receipt (digital or hard-copy) to the research team to get 50% of the cost reimbursed. After reimbursement, the total estimated cost to a participant who started PrEP was from 590 CNY [90 USD] for one-month generic PrEP supply or 30 pills ([200 mg emtricitabine, 300 mg tenofovir diso-proxil (fumarate)] [90 USD/month]) to 2000 CNY (307 USD) for a two-month supply of Tru-vada ([200 mg emtricitabine, 300 mg tenofovir disoproxil (fumarate)] [153 USD/month]). Once starting PrEP, participants were required to complete two monthly clinic visits during their first two months of PrEP use to monitor their medication adherence, HIV/STI status, and overall physical health status. Participants who decided to start or continue PrEP after Week 8 would still be able to receive standard PrEP care at the study hospital, but the expense for lab tests and PrEP medications was no longer reimbursed. All participants were informed during their last virtual study visit that another PrEP demonstration project at the study-affili-ated hospital would be open for enrollment in late 2021, where they could continue accessing PrEP.

## Assessments

We adopted a mixed-methods experimental (intervention) design [21] to evaluate the PrEP mini-app such that we embedded qualitative progress interviews before and during RCT, and combined the two types of data in constructing and interpreting results. Quantitative assess-ments were administered via web-based surveys at baseline and the 4th, 8th, and 12th weeks. Participants in the intervention arm were also required to fill out a short survey on self-reported mini-app use at the 2nd and 6th weeks. Qualitative in-depth interviews (up to two per participant) were conducted via WeChat call with a subgroup of participants from both arms at their 4th and 8th study weeks. These participants were purposively sampled for PrEP initia-tion status (yes vs. no) during the trial and self-reported mini-app use (high vs. low). Interview questions focused on three domains: (1) experiences using the intervention mini-app; (2) changes in perceptions and behaviors related to PrEP and other HIV prevention practices; and (3) general comments and feedback about the study. During the trial, only the HIV physician at the study clinic (LL) had access to personally identifiable information of participants who completed initial consultation and laboratory tests for PrEP initiation. After the conclusion of data collection, the research team had no access to information that can identify individual participants.

**Outcome measures.**   Quantitative outcome measures analyzed in this study include GBMSM's PrEP-related psychosocial and behavioral determinants, including intention to use PrEP, actual PrEP initiation, PrEP use self-efficacy, PrEP-related stigma and attitudes, and stages of change towards PrEP initiation. Definitions of these variables and assessment time-points are summarized in Table 1. For participants in the mini-app intervention arm, we also measured their self-reported mini-app engagement via survey, and the aggregated number of daily visits to the mini-app and HIV/syphilis test orders from the WeChat backend data. The self-reported mini-app engagement included frequency of using the mini-app on weekly aver-age (> 3 times/week, 1–2 times/week, < 1 time/week, and did not use the mini-app), perceived

**Table 1. Definitions and assessment timepoints of PrEP-related outcome variables.**

| Variable | Assessment timepoints | Definitions |
|---|---|---|
| PrEP initiation | Throughout the study course | Binary variable (1 = Yes, 0 = No)<br>This variable is not time-specified. |
| Stages of change towards PrEP initiation [23] | 0,4,8,12 weeks | Ordinal categorical variable (range 1–4):<br>1 = Pre-contemplation: No plan to get PrEP prescription within the next 3 months;<br>2 = Contemplation: Plan to get PrEP within the next 3 months;<br>3 = Preparation: Have discussed with doctors or informal PrEP providers about starting PrEP, and/or have completed initial lab tests for PrEP prescription; 4 = Action: Have obtained PrEP medication, and/or have started using PrEP |
| PrEP intention [24] | 0,4,8,12 weeks | Continuous variable (range -3 to 3): Response score on a single-item 7-point Bipolar scale (-3 = very unlikely to start PrEP at present, 3 = very likely to start PrEP at present). Higher score = higher intention to start PrEP. |
| PrEP self-efficacy [25] | 0,4,8,12 weeks | Continuous variable (range 1–5): Averaged response score on an 8-item scale with 5-point rating (1 = very difficult, 5 = very easy). Higher score = higher self-efficacy to use PrEP. |
| PrEP stigma [25] | 0,4,8,12 weeks | Continuous variable (range 1–5): Averaged response score on a 5-item scale with 5-point rating (1 = strongly disagree, 5 = strongly agree). Higher score = higher PrEP stigma. |
| PrEP attitudes [25] | 0,4,8,12 weeks | Continuous variable (range 1–5) Averaged response score on a 5-item scale with 5-point rating (1 = strongly disagree, 5 = strongly agree). Higher score = more positive attitudes towards PrEP. |

helpfulness of the mini-app, and perceived most helpful mini-app feature. At Week 8, participants were asked to rate the usability of the mini-app via the 10-item System Usability Scale (SUS) [22].

**Data analytic strategies.** Descriptive analyses were conducted to report the frequencies, means, and standard deviations (SDs) of participants' demographic characteristics, HIV- and PrEP-related psychosocial and behavioral variables, and self-reported use of the PrEP mini-app. Standardized differences [26] were calculated to compare baseline characteristics between the mini-app and the control groups. For the observed effect sizes of continuous outcome variables, measures of central tendency (means, medians, SDs) and Cohen's d with 95% confidence intervals (CI) were calculated. Cohen's d compared the net changes of continuous outcome variables from baseline to each follow-up between the two groups. For categorical outcome variables (e.g., PrEP initiation), Chi-Square test or Fisher's exact test was applied to compare the difference of distribution of outcome events between the two groups instead of relative risks, as the latter would be severely biased due to the small sample size [27,28]. To assess the mean changes between baseline and each follow-up assessment within each group (i.e., the mini-app group and the control group), we employed the Wilcoxon signed-rank test. To examine the between-group differences, we employed the Wilcoxon rank-sum test to investigate whether there were significant differences in the mean changes over time between the two groups. All statistical analyses were conducted using Stata software version 15.0 for Mac (StataCorp LLC, College Station, Texas, US).

All qualitative data from the progress evaluation interviews were audio-recorded, transcribed in Chinese, and analyzed in qualitative data analysis software NVivo 1.6 (QSR International Ltd, Melbourne, Australia). Thematic analysis using a primarily deductive approach [29] was implemented to understand participants' experience throughout the intervention period. A deductive codebook was first developed by CL based on the interview guide and applied to a random selection of five transcripts. The codebook was further revised by CL and KM adding emergent themes and inductive codes before application to all the transcripts. Analytical memos were utilized to note down reflections on the data during the coding process. CL took the lead role in analysis and the other research team members were actively engaged in the monitoring of the data collection process and providing feedback on data

analysis and interpretation. All qualitative analysis was conducted in Chinese with the translation of illustrative quotes for English language publications.

**Ethical approval.** This study and its protocols were reviewed and approved by the Institutional Review Boards of the University of North Carolina at Chapel Hill, USA (#19–3481), the Guangdong Provincial Dermatology Hospital, China (#2020031), and the Guangzhou Eighth People's Hospital, China (#202022155). During the eligibility screening, all participants provided verbal consent for screening to the researcher (CL) during the virtual enrollment visit via WeChat. Eligible participants were then proceeded to the formal informed consent process where they were instructed to read a digital consent form, given the opportunity to ask questions before finally electronically signing the informed consent through a digital survey platform. All signed consent forms were then downloaded to a password-protected local drive and stored separately from participants' survey and clinical visit data.

## Results

### Participants' demographic characteristics and study retention

Between 16 November 2020 and 2 April 2021, 118 people signed up for initial eligibility screening, among whom 75 completed the virtual screening, and 70 GBMSM were successfully enrolled (age mean = 28.0, SD = 6.1). Participants' baseline demographic characteristics and psychosocial variables, and the standardized differences of these variables between two groups are presented in Tables 2 and 3. Forty-six and 24 participants were randomly assigned to the mini-app group and the controlled group at 2:1 ratio, respectively. Twelve (17.1%) participants reported ever having had sex with a partner they knew to be living with HIV. Fourteen (20%) participants reported having used PEP before. Over half of participants reported "often" or "always" feeling anxious about HIV (53.1%, n = 34), and perceived little social support regarding HIV prevention. There were no statistically significant baseline group differences on the abovementioned characteristics. However, participants in the mini-app group were less likely to reporting having had casual sex within the past month (47.8% vs 66.7%, p = 0.02). Regarding PrEP-related psychosocial variables, the mean scores for behavioral determinants of PrEP initiation (intention to use PrEP, PrEP use self-efficacy, PrEP stigma, and PrEP attitudes) at baseline were not statistically different between the mini-app and control groups.

Six (8.6%) participants across both arms dropped out of the study before Week 12. The overall attrition rates were 7.1% at Week 4, and 8.6% at Week 8 and Week 12, and were not statistically significantly different between the two study groups. There was no statistically significant difference in baseline characteristics between the retained and lost-to-follow-up participants, except participants who did not complete all study follow-up points had higher PrEP stigma than those retained (mean = 2.89 vs. 2.00, p = 0.04). During the study course, two (2.9%) participants tested HIV-positive before PrEP initiation (one by the lab tests required for PrEP prescription and the other by self-test), and were referred to the HIV antiretroviral treatment (ART) clinic at the study hospital with the participant's permission. Eighteen participants (12 from the mini-app group and 6 from the control) received qualitative interviews (S2 Table).

### PrEP initiation cascade

Participants with complete baseline and follow-up data (n = 64, 91.4%) were included in the PrEP care cascade analysis (Table 4). Twenty-four participants (39.1%) contacted the study team indicating interest in PrEP eligibility consultation. Twenty-two (34.4%) completed the initial consultation and lab tests, among whom all were screened as PrEP eligible except one tested HIV-positive. Thirteen (18.6%) participants obtained and filled their initial PrEP

**Table 2. Participant demographic characteristics at baseline (N = 70).**

| Characteristics | | Total | | Mini-app | | Control | | Standardized difference |
|---|---|---|---|---|---|---|---|---|
| | | N = 70 | % | n = 46 | % | n = 24 | % | |
| Age (range: 18–54), mean (SD) | | 28.0 (6.1) | | 28.2 (6.8) | | 27.5 (4.6) | | 0.10 |
| Sexual orientation | | | | | | | | |
| | Heterosexual | 1 | 1.4 | 0 | 0.0 | 1 | 4.2 | 0.54 |
| | Gay | 58 | 82.9 | 39 | 84.8 | 19 | 76.2 | |
| | Bisexual | 8 | 11.4 | 4 | 8.7 | 4 | 16.7 | |
| | Not listed | 2 | 2.9 | 2 | 4.3 | 0 | 0.0 | |
| | Prefer not to say | 1 | 1.4 | 1 | 2.2 | 0 | 0 | |
| Highest education | | | | | | | | |
| | Primary & below | 0 | 0 | 0 | 0 | 0 | 0 | 0.51 |
| | Middle school | 2 | 2.9 | 2 | 4.4 | 0 | 0.0 | |
| | High school | 4 | 5.7 | 2 | 4.4 | 2 | 8.3 | |
| | Vocational school | 1 | 1.4 | 1 | 2.2 | 0 | 0.0 | |
| | Three-year college | 10 | 14.3 | 5 | 10.9 | 5 | 20.8 | |
| | Bachelor's degree | 40 | 57.1 | 28 | 60.9 | 12 | 50.0 | |
| | Master's degree | 13 | 18.6 | 8 | 17.4 | 5 | 20.8 | |
| Employment status | | | | | | | | |
| | Student | 16 | 22.9 | 12 | 26.1 | 4 | 16.7 | 0.41 |
| | Self-employed | 3 | 4.3 | 2 | 4.4 | 1 | 4.2 | |
| | Fulltime employed | 46 | 65.7 | 28 | 60.9 | 18 | 75.0 | |
| | Part time | 3 | 4.3 | 2 | 4.4 | 1 | 4.2 | |
| | Unemployed | 2 | 2.9 | 2 | 4.4 | 0 | 0 | |
| Monthly income | | | | | | | | |
| | <1500 | 5 | 7.1 | 3 | 6.5 | 2 | 8.3 | 0.48 |
| | 1500–3000 | 9 | 12.9 | 6 | 13.0 | 3 | 12.5 | |
| | 3001–5000 | 11 | 15.7 | 9 | 19.6 | 2 | 8.3 | |
| | 5001–8000 | 16 | 22.9 | 12 | 26.1 | 4 | 16.7 | |
| | >8000 | 29 | 41.4 | 16 | 34.8 | 13 | 54.2 | |
| Having any medical insurance | | | | | | | | |
| | Yes | 64 | 91.4 | 41 | 89.1 | 23 | 95.8 | 0.26 |

Note: Standardized difference was calculated using the stddiff command [29], developed based on the methods proposed by Yang & Dalton [30].

prescription for the first month (30 pills), of whom 13 (4.3%) refilled for a second 30 pills. The control group had slightly higher but non-significant percentages of participants reaching each of the PrEP cascade stages as compared to the mini-app group.

Among the 22 participants who went through the clinical procedures to assess eligibility for PrEP, many emphasized the benefits of having the on-site study research assistant (RA) present during their first clinical visit. Some described how the RA helped reduce their anxiety and uneasiness in navigating the PrEP clinic; expressed that they would be less confident in navigating through the clinical procedure without the RA's help.

## PrEP using experiences among GBMSM who initiated PrEP

Eleven (84.6%) out of 13 participants who initiated PrEP indicated they were using event-driven PrEP and two (15.4%) were using daily dosing strategy. Participants in the mini-app group (n = 8) had a shorter length of time between study enrollment and initial PrEP

**Table 3. Participants' sexual behavior history and HIV-relevant psychosocial characteristics at baseline.**

| Characteristics | | Total | | Mini-app | | Control | | Standardized difference |
|---|---|---|---|---|---|---|---|---|
| | | N = 70 | % | n = 46 | % | n = 24 | % | |
| *Sexual behavior* | | | | | | | | |
| Condom use during sex (lifetime) | | | | | | | | |
| | Every time | 19 | 27.1 | 14 | 30.4 | 5 | 20.8 | 0.68 |
| | Most of the time | 36 | 51.4 | 22 | 47.8 | 14 | 58.3 | |
| | Half of the time | 9 | 12.9 | 4 | 8.7 | 5 | 20.8 | |
| | Occasionally | 6 | 8.6 | 6 | 13.0 | 0 | 0.0 | |
| | Never | 0 | 0.0 | 0 | 0.0 | 0 | 0.0 | |
| Relationship status | | | | | | | | |
| | Single | 40 | 57.1 | 23 | 50.0 | 17 | 70.8 | 0.53 |
| | Not married, have male partner(s) | 22 | 31.4 | 18 | 39.1 | 4 | 16.7 | |
| | Married to a woman, have male partner(s) | 2 | 2.9 | 1 | 2.2 | 1 | 4.2 | |
| | Married to a woman, no male partner(s) | 3 | 4.3 | 2 | 4.4 | 1 | 4.2 | |
| | Not sure | 3 | 4.3 | 2 | 4.4 | 1 | 4.2 | |
| Sex with a partner living with HIV (lifetime) | | | | | | | | |
| | Yes | 12 | 17.1 | 9 | 19.6 | 3 | 12.5 | 0.19 |
| | No | 44 | 62.9 | 28 | 60.8 | 16 | 66.7 | |
| | Not sure | 14 | 20.0 | 9 | 19.6 | 5 | 20.8 | |
| Any casual sex partners (past month) | | | | | | | | |
| | Yes | 38 | 54.3 | 22 | 47.8 | 16 | 66.7 | 0.69 |
| | No | 30 | 42.9 | 24 | 52.2 | 6 | 25.0 | |
| | Not remember | 2 | 2.9 | 0 | 0 | 2 | 8.3 | |
| Number of sex partners (past 3 months) | | | | | | | | |
| | 0 | 2 | 2.9 | 2 | 4.4 | 0 | 0.0 | 0.54 |
| | 1 | 17 | 24.3 | 11 | 23.9 | 6 | 25.0 | |
| | 2–5 | 40 | 57.1 | 24 | 52.2 | 16 | 66.7 | |
| | 5–10 | 6 | 8.6 | 5 | 10.9 | 1 | 4.2 | |
| | 10–20 | 2 | 2.9 | 2 | 4.4 | 0 | 0.0 | |
| | >20 | 0 | 0 | 0 | 0 | 0 | 0 | |
| | Do not recall | 3 | 4.3 | 2 | 4.4 | 1 | 4.2 | |
| *HIV-related constructs* | | | | | | | | |
| Time since last HIV test | | | | | | | | |
| | Within last 3 months | 59 | 84.3 | 40 | 87.0 | 19 | 79.2 | 0.46 |
| | 3–6 months before | 5 | 7.1 | 4 | 8.7 | 1 | 4.2 | |
| | 6–12 months before | 5 | 7.1 | 2 | 4.3 | 3 | 12.5 | |
| | Do not recall | 1 | 1.5 | 0 | 0.0 | 1 | 4.2 | |
| HIV risk (Compared to my peers, "my" risk of getting HIV is) | | | | | | | | |
| | Little to none | 18 | 25.7 | 15 | 32.6 | 3 | 12.5 | 0.75 |
| | Very small | 23 | 32.9 | 16 | 34.8 | 7 | 29.2 | |
| | Half and half | 19 | 27.1 | 8 | 17.4 | 11 | 45.8 | |
| | Relatively high | 9 | 12.9 | 6 | 13.0 | 3 | 12.5 | |
| | Very high | 1 | 1.4 | 1 | 2.2 | 0 | 0.0 | |
| Feel anxiety about HIV infection | | | | | | | | |
| | Never | 1 | 1.4 | 1 | 2.2 | 0 | 0 | 0.42 |
| | Sometimes | 32 | 45.7 | 19 | 41.3 | 13 | 54.2 | |
| | Half and half | 28 | 40.0 | 21 | 45.7 | 7 | 29.2 | |
| | Often | 9 | 12.9 | 5 | 10.9 | 4 | 16.7 | |

*(Continued)*

**Table 3.** (Continued)

| Characteristics | | | Total | | Mini-app | | Control | | Standardized difference |
|---|---|---|---|---|---|---|---|---|---|
| | | | N = 70 | % | n = 46 | % | n = 24 | % | |
| | | Always | 0 | 0 | 0 | 0 | 0 | 0 | |
| | HIV social support (range -2 to 2, -2 = very negative, 2 = very positive) | | | | | | | | |
| | | Mean (SD) | | 0.3 (0.4) | | 0.4 (0.3) | | 0.2 (0.5) | 0.32 |
| Ever used PEP | | | | | | | | | |
| | Yes | | 14 | 20.0 | 10 | 21.7 | 4 | 16.7 | 0.13 |
| PrEP-related | | | | | | | | | |
| | Intention to use PrEP (range -3 to 3, -3 = very unlikely, 3 = very likely) | | | | | | | | |
| | | Mean (SD) | | 1.0 (1.5) | | 0.9 (1.4) | | 1.0 (1.6) | 0.04 |
| | PrEP use self-efficacy (Ranges 1 to 5, 1 = very difficult, 5 = very easy) | | | | | | | | |
| | | Mean (SD) | | 3.5 (0.5) | | 3.6 (0.5) | | 3.4 (0.5) | 0.47 |
| | PrEP attitudes (ranges 1 to 5, 1 = very negative, 5 = very positive) | | | | | | | | |
| | | Mean (SD) | | 4.0 (0.7) | | 4.0 (0.8) | | 3.8 (0.5) | 0.32 |
| | PrEP stigma (ranges 1 to 5, higher value = more stigmatizing) | | | | | | | | |
| | | Mean (SD) | | 2.8 (0.9) | | 2.7 (0.9) | | 3.0 (0.9) | 0.30 |

Note: Standardized difference was calculated using the stddiff command [26], developed based on the methods proposed by Yang & Dalton [30].

consultation (lab tests) than those in the control group (n = 5) (median: 18 vs 36 days; mean: 26.6 vs 29.2 days), and a shorter period between study enrollment and obtaining the first PrEP prescription (median: 26 vs 53 days; mean: 32.4 vs 50.0 days). However, these subsamples were small and differences did not reach statistical significance. In qualitative interviews, the most commonly mentioned reasons for delays in completing the PrEP consultation or obtaining the initial PrEP prescription were busy work schedules or the challenge to find a time for the clinic visit. In particular, one participant was waiting to finish his PEP treatment before transitioning to PrEP.

Qualitative interviews also revealed that not all of the participants who obtained PrEP medications actually started using PrEP right away. The reasons for delaying actual PrEP initiation included wanting to save the PrEP pills for future use, or not expecting to have sexual encounters in the near future. As one participant described:

*I bought it (PrEP) because I think it's better to put it aside. It might be very inconvenient if, when you really need it (PrEP), at that time you don't have any and you don't know where to get it. Since the study is offering a discounted price, it's good to save some. (ID68, mini-app, PrEP initiation = yes, 26 yrs)*

**Table 4. PrEP initiation cascade extracted from medical records at the study clinic.**

| Steps in PrEP initiation cascade | Total (N = 64) | | Mini-app (n = 43) | | Control (n = 21) | |
|---|---|---|---|---|---|---|
| | n | % | n | % | n | % |
| Interest in initial visit | 24 | 37 | 14 | 33 | 10 | 48 |
| Initial visit (lab tests) | 21 | 33 | 13 | 30 | 8 | 38 |
| Initial prescription | 17 | 27 | 11 | 26 | 6 | 29 |
| First prescription filled | 13 | 20 | 8 | 19 | 5 | 24 |
| 1-month clinical follow-up | 5 | 8 | 3 | 7 | 2 | 10 |
| Second prescription filled | 2 | 3 | 2 | 1 | 2 | 10 |
| 2-month clinical follow-up | 0 | 0 | 0 | 0 | 0 | 0 |

Participants who had started using PrEP at the time of interview generally reported that PrEP helped bring them peace of mind. This effect combined with few–if any–noticeable side effects, boosted their confidence to continue to use PrEP in the future. One participant (daily PrEP user) explicitly indicated in the qualitative interview that he discontinued PrEP after the first 30 pills due to his own health concerns. No other participants who had started taking PrEP during the trial reported to the study team that they stopped using PrEP during the trial.

## Barriers to uptake among GBMSM who did not initiate PrEP

Overall, 81.4% of participants (n = 51) did not initiate PrEP (i.e. not obtained PrEP medication through this study) during the study. The most common four barriers to PrEP use identified by these participants in the Week 8 follow-up survey were "financial concerns of PrEP" (n = 32, 62.7%), "worries about PrEP medication side effects" (n = 20, 39.2%), "prefer to use other HIV prevention strategies (e.g., condom)" (n = 17, 33.3%), and "do not like taking any medication" (n = 15, 29.4%). In qualitative interviews with nine participants who never initiated PrEP (i.e. not obtained PrEP medication through this study), besides the aforementioned barriers, themes emerged related to logistical/procedural barriers and psychosocial barriers. The logistical barriers included the anticipated burden of ongoing PrEP care and the limited operating hours of the PrEP clinic at the study hospital. As stated by one participant:

*After starting this study, I learned a lot about the process (to get PrEP). If I really want to start it (PrEP), I may have to take on some extra burdens, inconvenient things, such as physical examinations, monitoring my liver and kidney functions, testing for HIV or other infections. It made me know that it (PrEP) is not an easy thing. (ID55, intervention, PrEP initiation = no, 24 yrs)*

Psychosocial barriers focused on stigma, including the perceived stigma surrounding patients at infectious disease specialty hospitals, as well as generalized HIV-related stigma within society. The current way that PrEP was provided was at odds with participants' desire to maintain privacy around their sexual identity, and avoid association with HIV.

*If I have to go to the hospital using my real identity for PrEP consultation and prescription, I don't think it's helpful. I won't consider using PrEP. (. . .) If I go to the hospital, a hospital that is special for hepatitis or HIV treatment, and bring home a bag with the hospital's logo on it, I don't think I can do it. (ID48, intervention, PrEP initiation = no, 30 yrs)*

## Observed changes in PrEP-related psychosocial variables

Among participants with complete data (n = 64), no significant differences were observed of net changes regarding PrEP self-efficacy, PrEP stigma and PrEP attitudes between the mini-app group and the control group, except for the net changes in PrEP self-efficacy from baseline to Week 12 such that the mini-app group had a significantly higher increase in PrEP self-efficacy from baseline to Week 12 than the control group (0.13 vs. 0.04, p<0.05; Table 5). Among participants who did not start PrEP through the study project (n = 51, 79.7%), paired difference tests indicated that intentions to use PrEP gradually increased from 0.61 at baseline to 1.10 at Week 4 (p<0.05), 0.90 at Week 8 (p>0.05), and 1.12 at Week 12 (p>0.05), respectively (Table 4). Similar increasing trends were observed in the mini-app and the control groups; however, the increases at Week 4 and Week 12 were statistically significant in the mini-app group, but not in the control group.

**Table 5. PrEP-related behavioral determinants from baseline to follow-ups, test for trend, and observed effect sizes.**

| Variables | | Mini-app | | | Control | | | $p^2$ | Cohen's $d^3$ | 95% C.I. |
|---|---|---|---|---|---|---|---|---|---|---|
| | | Mean | SD | $p^1$ | Mean | SD | $p^1$ | | | |
| Intention to use PrEP (n = 51; Ranges -3 to 3, -3 = very unlikely, 3 = very likely; Higher value = higher intention) | | | | | | | | | | |
| | Baseline | 0.69 | 1.47 | – | 0.44 | 1.50 | – | – | – | – |
| | Week 4 | 1.37 | 1.55 | 0.004 | 0.50 | 1.63 | 0.81 | 0.17 | 0.44 | (-0.16, 1.04) |
| | Week 8 | 1.03 | 1.46 | 0.22 | 0.63 | 1.67 | 0.69 | 0.78 | 0.10 | (-0.50, 0.69) |
| | Week 12[4] | 1.31 | 1.41 | 0.02 | 0.69 | 1.66 | 0.55 | 0.37 | 0.26 | (-0.33, 0.85) |
| PrEP self-efficacy (n = 64; Ranges 1 to 5, 1 = very difficult, 5 = very easy; Higher value = higher self-efficacy) | | | | | | | | | | |
| | Baseline | 3.62 | 0.51 | – | 3.34 | 0.53 | – | – | – | – |
| | Week 4[4] | 3.69 | 0.64 | 0.30 | 3.29 | 0.60 | 0.80 | 0.44 | 0.31 | (-0.22, 0.83) |
| | Week 8[4] | 3.60 | 0.62 | 0.52 | 3.26 | 0.61 | 0.73 | 0.98 | 0.13 | (-0.39, 0.65) |
| | Week 12 | 3.75 | 0.57 | 0.06 | 3.24 | 0.57 | 0.27 | 0.03 | 0.51 | (-0.02, 1.04) |
| PrEP attitudes (n = 64; Ranges 1 to 5, 1 = very negative, 5 = very positive; Higher value = more positive attitudes) | | | | | | | | | | |
| | Baseline | 4.07 | 0.76 | – | 3.85 | 0.50 | – | – | – | – |
| | Week 4[4,5] | 4.14 | 0.59 | 0.34 | 3.90 | 0.50 | 0.46 | 0.69 | -0.02 | (-0.50, 0.54) |
| | Week 8[4] | 4.06 | 0.60 | 0.92 | 3.62 | 0.89 | 0.40 | 0.38 | 0.29 | (-0.24, 0.81) |
| | Week 12[4,5] | 4.09 | 0.62 | 0.66 | 3.87 | 0.56 | 0.82 | 0.58 | -0.01 | (-0.52, 0.53) |
| PrEP stigma (n = 64; Ranges 1 to 5, 1 = very positive (not stigmatizing),5 = very negative (stigmatizing); Higher value = higher stigma) | | | | | | | | | | |
| | Baseline | 2.76 | 0.92 | – | 3.13 | 0.87 | – | – | – | – |
| | Week 4[4] | 2.74 | 0.84 | 0.78 | 3.12 | 0.96 | 0.85 | 0.72 | -0.01 | (-0.54, 0.51) |
| | Week 8[4,5] | 2.85 | 0.81 | 0.45 | 3.31 | 0.69 | 0.14 | 0.40 | -0.13 | (-0.65, 0.40) |
| | Week 12 | 2.73 | 0.85 | 1.00 | 3.24 | 0.78 | 0.89 | 0.85 | -0.18 | (-0.71, 0.34) |

Note

1. Difference test of paired data (mean changes between baseline and each follow-up assessment) using Wilcoxon signed-rank test.

2. Difference test of unmatched data (net changes from baseline to each follow-up assessment) using Wilcoxon rank-sum test.

3. Cohen's d compares the difference of net changes of outcome variables from baseline to each follow-up between two arms.

5. Violation of normality (Shapiro-Wilk test).

6. Violation of homogeneity (variance-comparison test)

The observed effect sizes measured by Cohen's d (Table 5) indicate that compared to the control group, the mini-app group had larger increase in PrEP intentions and PrEP self-efficacy, and larger decrease in PrEP stigma along the study course. For example, In the comparison of net increases of PrEP intention from baseline to Week 4 between groups, the observed effect size 0.40 means participants in the mini-app group had a greater increase in PrEP intention than those in the control group by 0.4 standardized deviation units. Similarly, in the comparison of net changes of PrEP stigma from baseline to Week 8, the observed effect size -0.13 means participants in the mini-app group had a greater decrease in PrEP stigma than those in the control group by 0.13 standardized deviation units.

## Use of the PrEP mini-app and qualitative feedback

All 43 participants in the PrEP mini-app used the app at least once following their enrollment. During the study course (November 16, 2020 –April 6, 2021), the mini-app received an average daily traffic of 4.2 visits overall (SD = 5.5), calculated using the WeChat mini-app backend data. A total of 13 (28.3%) participants in the mini-app arm used the HIV/syphilis at-home test ordering feature to order 39 test kits (no cap on the numbers of tests kits each participant could order). During the active intervention period, a total of 10 (21.7%) participants initiated 13 online chats. No participants reported any technical issues during the trial. Some

participants mentioned in qualitative interviews that they had shared the passcode to the mini-app with friends outside of the study when they were asked about PrEP or HIV-related questions, which might have led to an overestimation of the daily visits to the mini-app by real participants.

At the end of the active intervention period (Week 8), most participants rated the mini-app as "somewhat helpful" (54.5%) or "very helpful" (27.3%). About 48.9% of the participants in the mini-app group rated the PrEP mini-app with a score indicative of "above-average" usability as measured by the SUS (S3 Table). In qualitative interviews, participants noted that the mini-app was less engaging once they had browsed through all the informational materials in the Mini-classroom, and cited this as the primary reason for reduced use over time. As one participant said:

> *Speak of the mini-app . . . I think the mini-app is helpful. I answered this in the follow-up survey, asking me how I have used the mini-app. I used the HIV self-test kit and read some articles. (. . .) Once after I know all about this (PrEP), I don't use it often. If I suddenly forget about something (about PrEP) or not sure of something, I will go back (to the mini-app). Or when I am really sick or need to look up something. If others ask me (about PrEP), I would just refer them to the mini-app, or share some of the articles to them. (ID23, intervention, PrEP initiation = yes, 31 yrs)*

Participants in qualitative interviews also suggested future app refinement could include: (1) regular updates of local news, events (specifically for LGBTQIA communities), and HIV/PrEP-related research findings; (2) expanding content topics in the Mini-classroom (e.g., tips for better sex, anal health, men's health in general, and how to better incorporate PrEP into daily life routines, etc.); (3) enriching the media forms of educational materials, such as using short-videos or infographics instead of text-heavy articles; (4) adding a pill-taking history tracker; and (5) adding more communication features that allow the users to interact with each other, such as an anonymous forum, comments under each of the articles in the Mini-classroom.

## Discussion

This pilot RCT with 70 Chinese GBMSM shows that compared to standardized HIV prevention care, the WeChat-based PrEP mini-app exhibits modest yet non-significant potential to facilitate PrEP uptake. The mini-app shows promise in reducing delays within the PrEP care cascade, while also increasing PrEP use intentions and self-efficacy among the sample. It also fosters more positive attitudes toward PrEP, and addressing some aspects of PrEP-related stigma.

Overall, GBMSM sampled in this study evaluated the PrEP mini-app as acceptable and helpful in HIV prevention, but the lack of content updates and limited variety in how content was presented (i.e. primarily text-heavy articles) discouraged frequent use. Compared to standard HIV prevention care, the PrEP mini-app showed the most potential for behavioral change within the first four weeks of the intervention. The features of a centralized space to provide essential PrEP-related information (the Mini-classroom) and online appointment scheduling for the initial PrEP clinical counseling through the mini-app helped shorten participants' wait times from study enrollment to the initial PrEP visit and their first PrEP prescription. As participants in our study moved back and forth across different stages of change towards PrEP initiation, having continued access to tailored information at their fingertips and the ability to act on that information when they were ready are two strengths of this

mHealth intervention. In this regard, digital health interventions offer the ability for real-time assessment, ongoing evaluation of an individual's behavioral change progress, and personalized feedback with appropriate designs [26].

While about half of study participants indicated at baseline some level of interest or intention to start PrEP (i.e. the "Preparation" stage), less than 20% (n = 13) actually filled an initial PrEP prescription by the end of active intervention. Notably, only three people who started PrEP refilled their prescription during the study. Drop-off at this stage in the cascade could be partly explained by financial burden and the fact that most participants who started PrEP chose event-driven dosing strategy (11/13), such that they might not need to refill within the study course. In addition, GBMSM's perceived burden of ongoing PrEP clinical care and concerns over long-term cost and potential side effects likely also contributed to the attrition. A systematic review of global PrEP clinical trials and real-world studies showed that over 40% of PrEP users discontinued PrEP within 6 months [30]. The high cost in spite of the study discount, discontinued easy access to PrEP care once out of the supportive structure of research projects, lack of motivation for long-term use, and changes of risk perception regarding HIV infection may all lead to discontinuation of PrEP [30].

Our results also indicated that the PrEP mini-app did not significantly increase the number of participants who initiated PrEP compared to the control group. Potential explanations may be the structural or logistical barriers that include the financial cost of PrEP medicines even after reimbursement, required clinical procedures at the study hospital for PrEP initiation, which require structural changes to make PrEP more easily accessible [31,32]. One of the missed opportunities for PrEP uptake in our study was that the PrEP clinic in the study hospital was only open during normal work hours on Monday through Friday, which made it challenging for people to find a break from work to visit the clinic. In this context, the feasibility and potential of having flexible hours of clinic operation or opening weekend PrEP clinics, or expanding PrEP provision through community organizations or pharmacy stores, call for further investigation. Furthermore, the RA-led clinic navigation during the initial PrEP counseling and lab tests procedure at the study hospital was positively received, which indicates the importance of in-person or on-site component of PrEP intervention even in the digital health era. An on-site care navigator may help to decrease patients' anxiety of seeing an infectious diseases physician, build their confidence in navigating through the PrEP care system, and hence lead to PrEP initiation [33,34].

Our study has several limitations. First, the convenience sample enrolled in our study may have introduced self-selection bias to the results such that before exposure to any intervention, our participants were more interested or willing to use PrEP than the general GBMSM population. Second, we had a relatively small sample size that limited our statistical power to detect differences. Given the exploratory nature of a pilot design, the main goal of our study was to evaluate acceptability of the PrEP mini-app among target user communities, to identify barriers to implementation, and to obtain preliminary estimates of variance in outcome measures of interest to inform future larger scale trials. Third, we were not able to track individual participant's mini-app usage due to limited access to the mini-app backend data. We addressed this limitation by corroborating participants' self-reported mini-app usage with qualitative interviews and solicited their feedback on intervention improvement. However, future iterations of the PrEP mini-app will seek to include a more robust para-data capture system in order to more accurately measure intervention exposure and explore patterns of intervention use [35].

Our study also has several important implications for future research and policy reform in HIV prevention and care in China. First, the strong preference for event-driven PrEP among Chinese GBMSM that we found should be considered in the design of future mHealth

interventions in this setting to ensure that adherence and refill supportive features can be tailored for various dosing strategies. Second, future iterations of the PrEP mini-app development–as well as other mHealth research projects for PrEP promotion—may consider adding participants' suggested features, pill taking assistance or tracker, an interactive forum for users, and more timely updates to information about PrEP, HIV, and general sexual health. Third, further investigation into how to maximize the benefits of digital technologies in public health intervention for PrEP uptake is warranted. Provider training for both PrEP care provision and digital health-based practice is also critical [36–38]. Lastly, though generic PrEP is already available in China that drastically reduces the financial cost, anticipated stigma or concerns of identity disclosure is consistent throughout our participants' narratives about PrEP utilization. How to provide friendly PrEP care and expand access to PrEP via alternative provision models in addition to traditional clinical settings is highly desired and promising in facilitating broader PrEP expansion. This is particularly needed for GBMSM and other communities at high risk of HIV who are still highly stigmatized and already less engaged with the public health system.

## Conclusion

The demonstrated feasibility and acceptability of the PrEP mini-app, as well as its potential strengths and limitations in facilitating PrEP uptake in our study sample offer lessons for future scalable mHealth efforts. Providing comprehensive information on how PrEP works and how to access it in respectful and plain language is critical for participant engagement. Future improvements in such mHealth solutions may include diversifying the content presentation in engaging media formats (e.g., short videos), adding user engagement features (e.g., anonymous forums, comments under articles, live stream, ability to share components of the intervention with peers), and providing off-line in-clinic navigation support during initial PrEP visit. More importantly, structural barriers often pose a greater challenge to successful PrEP uptake than individual motivation and intention. Existing gaps in the shortage of PrEP providers and restricted channels for formal PrEP care are often posing a greater challenge need to be bridged to improve PrEP accessibility overall.

## Supporting information

**S1 Checklist. CONSORT checklist for randomized trial.**
(DOCX)

**S1 Table. Summary of the PrEP Mini-app key functions.**
(DOCX)

**S2 Table. Characteristics of interviewed participants.**
(DOCX)

**S3 Table. System usability scale scores.**
(DOCX)

**S1 File.**
(DOCX)

**S2 File.**
(PDF)

## Acknowledgments

We thank all the participants for sharing their experience and their time. We would also acknowledge the help from following individuals and organizations: Honglin Liu and Danyang Luo for their support and constructive feedback along the study process; the Guangzhou Zhitong (智同) LGBT Center and the Shenzhen Aitongxing (爱同行) LGBT Center for their assistance in study advertising and participants recruitment; Cheryl Hendrickson (University of North Carolina at Chapel Hill) and Dandan Gong (Guangzhou Eighth People's Hospital) for their assistance in study registration and IRB application.

## Author Contributions

**Conceptualization:** Chunyan Li, Suzanne Maman, Derrick D. Matthews, Edwin B. Fisher, Weiming Tang, Renslow Sherer, Aniruddha Hazra, Jonathan Lio, Linghua Li, Joseph D. Tucker, Kathryn E. Muessig.

**Data curation:** Chunyan Li, Yuan Xiong.

**Formal analysis:** Joseph D. Tucker, Kathryn E. Muessig.

**Funding acquisition:** Weiming Tang, Joseph D. Tucker.

**Investigation:** Chunyan Li, Linghua Li, Joseph D. Tucker.

**Methodology:** Joseph D. Tucker, Kathryn E. Muessig.

**Project administration:** Chunyan Li, Yuan Xiong, Haojie Huang, Linghua Li.

**Resources:** Linghua Li, Joseph D. Tucker, Kathryn E. Muessig.

**Software:** Tong Mu, Xiaokai Tong, Jianxiong Yu, Zeyu Yang.

**Supervision:** Suzanne Maman, Derrick D. Matthews, Edwin B. Fisher, Weiming Tang, Linghua Li, Joseph D. Tucker, Kathryn E. Muessig.

**Visualization:** Tong Mu, Xiaokai Tong.

**Writing – original draft:** Chunyan Li.

**Writing – review & editing:** Suzanne Maman, Derrick D. Matthews, Edwin B. Fisher, Weiming Tang, Haojie Huang, Renslow Sherer, Aniruddha Hazra, Jonathan Lio, Linghua Li, Joseph D. Tucker, Kathryn E. Muessig.

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
