## [Decision Letter · Decision Letter 0]

4 Jul 2023

PONE-D-23-10021An instant messaging mobile phone application for promoting HIV pre-exposure prophylaxis uptake among Chinese gay, bisexual and other men who have sex with men: A mixed methods feasibility and piloting randomized controlled trial studyPLOS ONE

Dear Dr. Li,

Thank you for submitting your manuscript to PLOS ONE. After careful consideration, we feel that it has merit but does not fully meet PLOS ONE’s publication criteria as it currently stands. Therefore, we invite you to submit a revised version of the manuscript that addresses the points raised during the review process.

Please submit track changes manuscript and clean version with responses to reviewers questions in letter . 

 Please submit your revised manuscript by Aug 18 2023 11:59PM. If you will need more time than this to complete your revisions, please reply to this message or contact the journal office at plosone@plos.org. Please include the following items when submitting your revised manuscript:A rebuttal letter that responds to each point raised by the academic editor and reviewer(s). You should upload this letter as a separate file labeled 'Response to Reviewers'.A marked-up copy of your manuscript that highlights changes made to the original version. You should upload this as a separate file labeled 'Revised Manuscript with Track Changes'.An unmarked version of your revised paper without tracked changes. You should upload this as a separate file labeled 'Manuscript'.If applicable, we recommend that you deposit your laboratory protocols in protocols.io to enhance the reproducibility of your results. Protocols.io assigns your protocol its own identifier (DOI) so that it can be cited independently in the future. For instructions see: https://journals.plos.org/plosone/s/submission-guidelines#loc-laboratory-protocols. Additionally, PLOS ONE offers an option for publishing peer-reviewed Lab Protocol articles, which describe protocols hosted on protocols.io. Read more information on sharing protocols at https://plos.org/protocols?utm_medium=editorial-email&utm_source=authorletters&utm_campaign=protocols.

We look forward to receiving your revised manuscript.

Kind regards,

Sandra Ann Springer, M.D.

Academic Editor

PLOS ONE

“This study was supported by the National Institute of Allergies and Infectious Diseases (Grant# R01-AI114310-S1, JT), National Institute of Mental Health (Grant # R34119963, JT & WT), National Natural Science Foundation of China (Grant#81903371, WT), the Graduate School of University of North Carolina at Chapel Hill (CL), and Tokyo College at the University of Tokyo (CL).  “

5. We noted in your submission details that a portion of your manuscript may have been presented or published elsewhere. [Figure 1 is taken from the study protocol that was published by BMJ Open (doi:10.1136/bmjopen-2021-055899). This is necessary because the figure depicts the user interface of the intervention studied.] Please clarify whether this publication was peer-reviewed and formally published. If this work was previously peer-reviewed and published, in the cover letter please provide the reason that this work does not constitute dual publication and should be included in the current manuscript.

6. We note that you have indicated that data from this study are available upon request. PLOS only allows data to be available upon request if there are legal or ethical restrictions on sharing data publicly. For information on unacceptable data access restrictions, please see http://journals.plos.org/plosone/s/data-availability#loc-unacceptable-data-access-restrictions.

7. Please ensure that you include a title page within your main document. You should list all authors and all affiliations as per our author instructions and clearly indicate the corresponding author.

8. We note that Figure 2 in your submission contain copyrighted images. All PLOS content is published under the Creative Commons Attribution License (CC BY 4.0), which means that the manuscript, images, and Supporting Information files will be freely available online, and any third party is permitted to access, download, copy, distribute, and use these materials in any way, even commercially, with proper attribution. For more information, see our copyright guidelines: http://journals.plos.org/plosone/s/licenses-and-copyright.

9. We note that the original protocol file you uploaded contains a confidentiality notice indicating that the protocol may not be shared publicly or be published. Please note, however, that the PLOS Editorial Policy requires that the original protocol be published alongside your manuscript in the event of acceptance. Please note that should your paper be accepted, all content including the protocol will be published under the Creative Commons Attribution (CC BY) 4.0 license, which means that it will be freely available online, and any third party is permitted to access, download, copy, distribute, and use these materials in any way, even commercially, with proper attribution.

Therefore, we ask that you please seek permission from the study sponsor or body imposing the restriction on sharing this document to publish this protocol under CC BY 4.0 if your work is accepted. We kindly ask that you upload a formal statement signed by an institutional representative clarifying whether you will be able to comply with this policy. Additionally, please upload a clean copy of the protocol with the confidentiality notice (and any copyrighted institutional logos or signatures) removed.

10. We note that the original protocol that you have uploaded as a Supporting Information file contains an institutional logo. As this logo is likely copyrighted, we ask that you please remove it from this file and upload an updated version upon resubmission.

Reviewers' comments:

Reviewer's Responses to Questions

**Comments to the Author**

1. Is the manuscript technically sound, and do the data support the conclusions?

Reviewer #1: Partly

Reviewer #2: Yes

2. Has the statistical analysis been performed appropriately and rigorously? 

Reviewer #1: Yes

Reviewer #2: Yes

3. Have the authors made all data underlying the findings in their manuscript fully available?

Reviewer #1: Yes

Reviewer #2: No

4. Is the manuscript presented in an intelligible fashion and written in standard English?

Reviewer #1: Yes

Reviewer #2: Yes

5. Review Comments to the Author

Reviewer #1: This is a well-written manuscript describing a study evaluating the utility of a mobile phone-based application for promoting HIV pre-exposure prophylaxis (PrEP) uptake among Chinese gay, bisexual and other men who have sex with men (GBMSM).

There are many strengths to the manuscript and the study described within the manuscript.

The app was designed to be culturally appropriate and GBMSM-friendly. Several steps were taken in the development process to ensure relevance to the target audience. First, the initial prototype of the PrEP education mini-app was developed through a Gay-Friendly Doctor Finder Hackathon. Additionally, the app was co-developed with community partners, HIV physicians, and sex education experts to develop the intervention content and user interface design. And early feedback sessions were conducted with GBMSM to test use the app in which participants gave verbal feedback during one-to-one in-depth interviews. The app was then refined based on this qualitative feedback.

Little research has been published on digital health applications with GBMSM in China and thus this study adds a novel contribution to the literature.

The mixed methods design of the randomized trial is a strength and a thorough array of outcome measures were included. Several limitations of the study are clearly stated.

The largest gap in this paper is that the content/tools/functionality of the app itself are not described in any detail in the methods section. Thus, the reader does not fully understand what the app actually does or how participants are advised to use it. The reader actually learns more about components of the app in the discussion section, where some examples of app content are referenced.

Also, the manuscript presents this study as if it is a parallel groups design in which participants are randomly assigned to either the intervention or control intervention; however, the study really is examining the utility of adding the app to standard care (as both groups received the control intervention). This should be presented more clearly in the manuscript.

The abstract should make it clear that no differences across groups were observed in key PrEP outcomes (those seeking consult for PrEP and those filling PrEP prescription). The abstract currently presents these findings collapsed across groups but is silent re: comparisons across conditions.

The rationale for randomizing participants in a 2:1 ratio in the intervention vs. control group is not provided.

The content in the Conclusions section of the abstract appears to overstate the promise of the app, given that the preceding Results section in the abstract does not clearly reflect added benefit of the app.

Overall, although this manuscript is well-written, it is lengthy and could be presented a bit more efficiently.

Reviewer #2: General comments:

From a statistical and study design perspective, this manuscript largely looks good. I had some comments on specific lines and tables of the manuscript that are meant to improve the quality of the manuscript.

I also have a couple general comments. First, you make it clear that this pilot study and I think the presentation and interpretation of the results mostly concurs with this type of design. One piece that you may want to think about is about how you present statistical significance. My feeling here is that the statistical significance does not matter much here given that this study is trying to establish conceptual proof that this intervention could work. Thus, really what is most important is the direction and (potentially) magnitude of the effects. It's clear that this study is underpowered, but that's fully understandable. Therefore the confidence intervals and p-values do not mean as much as they would in a larger study that endeavors to confirm a concept (though I think it's still think it's worthwhile to some to report them).

Second, regarding the magnitude of the effects, I think it would be helpful if you provided some information on how much higher the effects are. For instance, in table 5, for intention to use PrEP, d=0.44 for week 4. What does that 0.44 indicate? Although Cohen's d is a well-accepted measure in the social sciences, it is less well-known in medicine and public health. You might consider comparisons on the raw scale to aid interpretation, but I will leave this up to you. Regardless, I would not short sell the importance of the effect sizes here. Although this is a small study, those effect sizes are important for future research.

Specific comments:

1. (line 72) "rea" should be "area".

2. (Tables 2 and 3) Significance testing between intervention groups is generally frowned upon because a non-significant p-value does not indicate that groups are the same. For info on the topic in relation to baseline imbalance in randomized trials see Altman, https://doi.org/10.2307/2987510 and Senn, https://doi.org/10.1002/sim.4780131703. My recommendation is to remove the significance testing from tables like this and use standardized difference to assess differences (see Austin, https://doi.org/10.1080/03610910902859574).

3. (Table 5) Please mention all these tests (e.g., Wilcoxon, Shapiro-Wilk's) in the statistical methods section.

4. (line 279) I don't know that I would call these "trends" since I don't think changes over multiple time points are being assessed here. Does "differences" work here?

5. (lines 336-338) I think you've stated why you are using Cohen's d in the methods, so I don't feel like you need to restate it here. Maybe if you feel the description in the methods is insufficient, you can move this statement to that section.

6. (p.26, lines 36-39) I suggest cutting the first two sentences of the discussion. These sentences seem to be restating the aims of the study and take away from the main conclusions of the manuscript.

6. PLOS authors have the option to publish the peer review history of their article (what does this mean?). If published, this will include your full peer review and any attached files.

Reviewer #1: No

Reviewer #2: No

---

## [Author Response · Author response to Decision Letter 0]

3 Aug 2023

Dear Dr. Sarah Ann Springer and anonymous reviewers,

Thank you for reviewing our manuscript. We appreciate the constructive comments from the editor and three peer reviewers to further strengthen our paper. For the editor’s comments regarding journal requirements, we included our responses in the revised cover letter. Below is our point-by-point response to the reviewers’ comments, grouped by manuscript section.

Sincerely,

Chunyan Li & Joseph D. Tucker, on behalf of the author team

Point-by-point response

Overall:

Reviewer #1: “Overall, although this manuscript is well-written, it is lengthy and could be presented a bit more efficiently.”

Response: We appreciate the reviewer’s comment on our study presentation and agree this is a slightly long manuscript. As this is a mixed-methods design that contains both a RCT and qualitative progress evaluation component, we tried to present a comprehensive overview of the study findings to inform future PrEP intervention research. In this round of revision, we removed redundant words and clarified many sentences.

Abstract: 

Reviewer #1: “The abstract should make it clear that no differences across groups were observed in key PrEP outcomes (those seeking consult for PrEP and those filling PrEP prescription). The abstract currently presents these findings collapsed across groups but is silent re: comparisons across conditions.”

Response: In the Results section of the Abstract, we included a sentence as “We observed modest but non-significant improvements in participants’ intention to use PrEP, actual PrEP initiation, PrEP-related self-efficacy, stigma, and attitudes over 12 weeks when comparing the mini-app and the control arms” to articulate the non-significance across treatment groups. In the Conclusion, we also highlighted that this is a pilot trial with a small sample size that primarily aimed to demonstrate feasibility and preliminary evidence of efficacy. 

Methods:

Reviewer #1: “The largest gap in this paper is that the content/tools/functionality of the app itself are not described in any detail in the methods section. Thus, the reader does not fully understand what the app actually does or how participants are advised to use it. The reader actually learns more about components of the app in the discussion section, where some examples of app content are referenced.”

Response: Thank you for pointing this out. We added a brief description of the 4 key features of the PrEP mini-app in the methods section (Page 7, line 164-178), and made it clear that a full description of the mini-app function and its theoretical foundations can be found in the supplementary file. Figure 2 also presents the screenshots of the mini-app, presenting its core functions (Page 7). The development process of the mini-app is published in the trial protocol (BMJ Open 2022;12:e055899. doi:10.1136/ bmjopen-2021-055899) which is cited in the intervention description section as well. 

Reviewer #1: “The manuscript presents this study as if it is a parallel groups design in which participants are randomly assigned to either the intervention or control intervention; however, the study really is examining the utility of adding the app to standard care (as both groups received the control intervention). This should be presented more clearly in the manuscript.”

Response: We apologize for the confusion. In the revised version, we clarified that participants in the RCT were “randomly assigned to the mini-app group (i.e. the intervention app plus standard care) or the control group (i.e. standard care only) at a 2:1 ratio using a permuted block randomization” (Page 6, line 134-136). We also updated the CONSORT diagram accordingly. 

Reviewer #1: “The rationale for randomizing participants in a 2:1 ratio in the intervention vs. control group is not provided.”

Response: The rationale behind 2:1 allocation is now added on Page 6 line 136-138 as: “The 2:1 allocation was used to ensure the intervention arm had enough samples to capture the range of users’ reactions to the mini-app and its content.” 

Reviewer #2: “(Table 5) Please mention all these tests (e.g., Wilcoxon, Shapiro-Wilk's) in the statistical methods section.”

Response: We have expanded the methods description in text accordingly. (Page 11, line 264-268)

Results:

Reviewer #2: “I also have a couple general comments. First, you make it clear that this pilot study and I think the presentation and interpretation of the results mostly concurs with this type of design. One piece that you may want to think about is about how you present statistical significance. My feeling here is that the statistical significance does not matter much here given that this study is trying to establish conceptual proof that this intervention could work. Thus, really what is most important is the direction and (potentially) magnitude of the effects. It's clear that this study is underpowered, but that's fully understandable. Therefore the confidence intervals and p-values do not mean as much as they would in a larger study that endeavors to confirm a concept (though I think it's still think it's worthwhile to some to report them).”

Response: We appreciate the reviewer’s comment on the limitations of a pilot study design due to its exploratory nature. We agree that the magnitude and direction of potential effect sizes (or difference between treatment groups) are more important than the statistical significance test. We decided to report the p-values and confidence intervals in Table 5 for consistency and transparency, and provide readers with a useful reference point for better interpretation of the study's outcomes.

Reviewer #2: “Second, regarding the magnitude of the effects, I think it would be helpful if you provided some information on how much higher the effects are. For instance, in table 5, for intention to use PrEP, d=0.44 for week 4. What does that 0.44 indicate? Although Cohen's d is a well-accepted measure in the social sciences, it is less well-known in medicine and public health. You might consider comparisons on the raw scale to aid interpretation, but I will leave this up to you. Regardless, I would not short sell the importance of the effect sizes here. Although this is a small study, those effect sizes are important for future research.”

Response: We appreciate the reviewer’s suggestion and agree that Cohen’s d might be less familiar to some readers. In future studies, we will try to include comparison on raw scales to make it more accessible to all readers. In Table 5, we reported the raw scale of each PrEP-related variables at all time points to aid data interpretation. Additionally, in the revised manuscript (Page 24, line 610-616), we gave two examples of how to interpret the magnitude and direction of the effect sizes. 

“For example, In the comparison of net increases of PrEP intention from baseline to Week 4 between groups, the observed effect size 0.40 means participants in the mini-app group had a greater increase in PrEP intention than those in the control group by 0.4 standardized deviation units. Similarly, in the comparison of net changes of PrEP stigma from baseline to Week 8, the observed effect size -0.13 means participants in the mini-app group had a greater decrease in PrEP stigma than those in the control group by 0.13 standardized deviation units”. 

Reviewer #2: “(Tables 2 and 3) Significance testing between intervention groups is generally frowned upon because a non-significant p-value does not indicate that groups are the same. For info on the topic in relation to baseline imbalance in randomized trials see Altman, https://doi.org/10.2307/2987510 and Senn, https://doi.org/10.1002/sim.4780131703. My recommendation is to remove the significance testing from tables like this and use standardized difference to assess differences (see Austin, https://doi.org/10.1080/03610910902859574).”

Response: Thank you for the suggestion of reporting standardized difference of variables between treatment groups instead of p-values. The references above are also very helpful. In the revised version, we used the stddiff command in Stata to calculate the standardized differences in Tables 2-3. The stddiff command was developed by Ahmed M. Bayoumi (https://EconPapers.repec.org/repec:boc:bocode:s458275) based on the methods proposed by Yang & Dalton (http://support.sas.com/resources/papers/proceedings12/335-2012.pdf). The results description was updated accordingly to reflect the inclusion of standardized differences. 

Reviewer #2: “(line 279) I don't know that I would call these "trends" since I don't think changes over multiple time points are being assessed here. Does "differences" work here?”

Response: Thank you for the suggestion. We replaced “trends” with “differences” in the revised manuscript. 

Reviewer #2: “I think you've stated why you are using Cohen's d in the methods, so I don't feel like you need to restate it here. Maybe if you feel the description in the methods is insufficient, you can move this statement to that section.”

Response: Thank you for the suggestion. We believe we have provided enough information about the use of Cohen’s d in the Methods section. The redundant description of Cohen’s d in the Results section is now removed. 

Discussion:

Reviewer #2: “I suggest cutting the first two sentences of the discussion. These sentences seem to be restating the aims of the study and take away from the main conclusions of the manuscript.”

Response: We appreciate the advice. The two sentences are now removed and the paragraph is edited accordingly for readability.

---

## [Editor Report · Decision Letter 1]

30 Oct 2023

An instant messaging mobile phone application for promoting HIV pre-exposure prophylaxis uptake among Chinese gay, bisexual and other men who have sex with men: A mixed methods feasibility and piloting randomized controlled trial study

PONE-D-23-10021R1

Dear Dr. Li,

We’re pleased to inform you that your manuscript has been judged scientifically suitable for publication and will be formally accepted for publication once it meets all outstanding technical requirements.

Kind regards,

Sandra Ann Springer, M.D.

Academic Editor

PLOS ONE
---

## [Editor Report · Acceptance letter]

5 Nov 2023

PONE-D-23-10021R1 

An instant messaging mobile phone application for promoting HIV pre-exposure prophylaxis uptake among Chinese gay, bisexual and other men who have sex with men: A mixed methods feasibility and piloting randomized controlled trial study 

Dear Dr. Li:

I'm pleased to inform you that your manuscript has been deemed suitable for publication in PLOS ONE. Congratulations! Your manuscript is now with our production department. 

Kind regards, 

on behalf of

Dr. Sandra Ann Springer 

%CORR_ED_EDITOR_ROLE%

PLOS ONE